# Prevalence and Sociodemographic Profiles of Grand Multipara in Abu Dhabi, United Arab Emirates

**DOI:** 10.3390/nu14214686

**Published:** 2022-11-05

**Authors:** Zainab Taha, Farid El Ktaibi, Aysha Ibrahim Al Dhaheri, Dimitrios Papandreou, Ahmed Ali Hassan

**Affiliations:** 1Department of Health Sciences, College of Natural and Health Sciences, Zayed University, Abu Dhabi P.O. Box 144534, United Arab Emirates; 2Department of Mathematics and Statistics, College of Natural and Health Sciences, Zayed University, Abu Dhabi P.O. Box 144534, United Arab Emirates; 3Institute of Public Health, College of Medicine & Health Science, Abu Dhabi P.O. Box 15551, United Arab Emirates; 4Department of Research, Taami for Agricultural and Animal Production, Khartoum, Sudan

**Keywords:** grand multipara, Arab mothers, maternal age, overweight, United Arab Emirates

## Abstract

The literature shows that grand multipara mothers are major contributors to poor maternal and perinatal health compared to multipara mothers. Data regarding parity profiles are essential, especially in rapidly transforming countries such as the United Arab Emirates (UAE). This study aimed to investigate the prevalence and factors associated with multipara mothers compared to multipara mothers in Abu Dhabi, UAE. The data were collected from seven health care centers located in Abu Dhabi.From1818 enrolled mothers, the prevalence of grand multipara was 135(7.4%, 95% Confidence Interval (CI) = 7.2, 7.6). In logistic regression analysis, factors associated positively with grand multipara were a higher maternal age (Adjusted Odd Ratio (AOR) = 1.28, 95% CI = 1.21, 1.34), Arab mothers (AOR = 5.66, 95% CI 2.81, 11.40), overweight pre-pregnancy (AOR = 2.01, 95% CI = 1.26, 3.21), and limited family support for breastfeeding (AOR = 2.05, 95% CI = 1.21, 3.50). The prevalence of grand multipara was low compared to previous researching the UAE. Sociodemographic factors were more prominent and associated with grand multipara mothers compared to obstetrical ones. Therefore, more programs (nutritional, physical activities, and psychosocial) are needed to improve maternal and perinatal health to support grand multipara mothers. Further, research is required to explore the difference in parity based on nationalities, especially from a sociocultural point of view.

## 1. Introduction

Grand multipara mothers (five or more births) are at high risk of poor maternal and perinatal health, such as stillbirth, preterm birth delivery [1,2], low birth weight (LBW), gestational hypertension, admission to neonatal intensive care unit (NICU) [1], anemia [3], diabetes mellitus [1,3,4], and cesarean section (CS) [1,3] compared to low-parity mothers. Poor health associated with grand multipara is more common in low- and middle-income countries (LMICs) [3,5,6]. Such poor health in LMICs can be attributed to many reasons, such as low maternal education; poor access to health care services, including family planning; lack of antenatal and perinatal care; and poor maternal nutrition [3,5,6]. Fortunately, these issues are manageable through developing and implementing appropriate programs with a solid commitment from all involved parties. 

In the United Arab Emirates (UAE), the high admissions of grand multipara babies to the NICU was attributed to a high prevalence of diabetes mellitus among grand multipara compared to multipara women [4]. Unlike LMICs, such as Sudan [7], admissions to NICUs in the UAE are associated with better outcomes in terms of morbidity and mortality due to the provision of high-quality health care [8]. The lack of association between grand multipara and poor obstetrics outcomes in the UAE and Saudi Arabia could be explained by the existence of high-quality care in these countries [4,9]. 

Data from the literature have shown that increased family planning is a possible contributor to the low prevalence of grand multipara in Western countries. However, this phenomenon has started shifting to other countries, and the UAE is no exception [5,10]. The debate is ongoing in the literature about best practices and whether it is better to have fewer or many children in LMICs versus high-income countries (HICs) [11,12]. Unlike in LMICs [5], providing free and high-quality health care could encourage parents to have many children without resulting in poor maternal and perinatal outcomes [9]. Different prevalence rates of grand multipara have been reported by researchers in many countries, including the UAE(36.6%) in 2001 [4] and Jordan (1.8%) [1] and Ethiopia (70.8%) inthe demographic and health survey of 2000–2016 [13]. Many factors affect birth rates indifferent countries, such as social structure, religious beliefs, economic status, and urbanization [11]. Parity is a complex issue as it is linked to human fertility. In addition, human fertility is related to education, the economy, religion, contraception, and family planning programs [14]. In the presence of contradictory data about grand multipara issues (i.e., the acceptance of having more children and the fear of pregnancy complications), it is essential for each country to have its own updated and conclusive data on grand multipara and its impact on pregnancy and neonatal outcomes.

To provide decision-makers with reliable information to develop strategic plans aiming to improve grand multipara women’s health and pregnancy outcomes, first, the prevalence of grand multipara needs to be estimated; second, the impact of grand multipara on poor maternal and perinatal health needs to be highlighted; and third, factors associated with grand multipara, including sociocultural factors, need to be addressed. Unfortunately, data regarding grand multipara in the UAE are scarce [4].

Therefore, updated data regarding the parity profile is essential, especially in a country characterized by rapid economic and cultural transformation, such as the UAE [10,15]. Providing data about grand multipara is of paramount importance to help policymakers from different ministries, including the Ministry of Health, adopt appropriate strategies to tackle such important issues. This study aimed to investigate the prevalence and factors associated with grand multipara mothers compared to multipara mothers in Abu Dhabi, the UAE.

## 2. Materials and Methods

### 2.1. Study Design and Setting 

A multicenter cross-sectional study was conducted at seven health care centers located in the urban, suburban, and rural areas of Abu Dhabi from March to September 2017. The current data are part of a large project aimed to improving maternal and child health in Abu Dhabi, UAE. Abu Dhabi is the capital of the UAE and the largest emirate among the seven emirates of the UAE. Based on the recent estimation of the World Bank, about 10 million people live in the UAE [16]. The project’s data were collected from both Emirati and non-Emirati mothers. 

### 2.2. Outcome Measures 

The primary outcome measured in this study was parity (grand multipara vs. multipara). 

Secondary outcomes were factors associated with grand multipara, such as sociodemographic factors (e.g., age, parent education, and employment) and obstetrical factors (e.g., parity, mode of delivery, gestational age, and weight at delivery).

### 2.3. Definitions

Parity: number of times a woman has given birth to a fetus with a gestational age of ≥22 weeks, regardless of whether the child was born alive or stillborn. This is further categorized into grand multipara mothers (five or more births) and multipara mothers (2 to 4 births) [4].

Body Mass Index (BMI): calculated as the weight in kilograms divided by the square of the height in meters (kg/m^2^) [17]. This information (pre-pregnancy maternal height and weight) was extracted from a maternal health card. The BMI was categorized into subgroups using the WHO classification as follows: 

Underweight (BMI of <18.5 Kg/m^2^);

Normal weight (BMI of 18.5–24.9 Kg/m^2^);

Overweight (BMI of 25–29.9 Kg/m^2^);

Obese (BMI of ≥30 Kg/m^2^)

Family support: this information was obtained by asking the mother about any kind of support and encouragement she received, including breastfeeding, from family members, relatives, and other non-relatives.

Preterm birth is the birth of a baby at <37 weeks gestational age at delivery.

Low birth weight is defined as the weight of the newborn infant of <2500 g immediately after delivery.

Arab nationality: this category included all Emirati and non-Emirati mothers.

Non-Arab nationality: this category included Asian mothers and mothers from other nationalities.

Cesarean Section: a surgical procedure in which incisions are made through a woman’s abdomen and uterus to deliver her baby.

### 2.4. Data Collection

The main tool for the current data collection was a structured questionnaire. The questionnaire was tested first, and every error was corrected before data collection. The questionnaire included family demographics (e.g., education, age, nationality, occupation), child’s information (e.g., birth weight and height, gestational age at delivery), and maternal obstetrical information (e.g., parity and mode of delivery). Female research assistants were trained by the investigators to collect the data using the questionnaire and interviewing mothers after ensuring all ethical issues were fulfilled. The ethical issues include the following: the right to participate and the right to withdraw at any time during the study, the right to ask questions regarding the study, and the privacy of the collected data. More information about the study’s design is given elsewhere [18]. The study was approved by the ethics committee of Zayed University (R17042), and all participants signed a consent form.

### 2.5. Study Population

All mothers (Emirati and non-Emirati mothers) who came to health care centers were approached by trained medical assistants during the study period. The purpose of the study and the ethical issues were explained to the eligible mothers by the medical assistants. 

### 2.6. Study Inclusion and Exclusion Criteria 

During the study period, a total of 1822 mother–child pairs were enrolled in the project. The data with completed and interesting outcomes were included in the analysis, such as parity and mode of delivery. In this paper, primipara mothers were excluded from the analysis, assuming they have special obstetric categories, similar to a previous study [4].

### 2.7. Statistical Analysis

Mothers with 5 deliveries were considered grand multipara and coded as (1), and mothers with 2 to 4 deliveries were considered multipara and coded as (0). Initially, the data were entered into an Excel file and then transformed into the statistical package for the social sciences (SPSS) version 22 and double-checked before analysis. Descriptive statistics, as well as inferential statistics, were applied to analyze the data. Continuous data such as maternal age were checked for normality using the Shapiro–Wilk test. Next, the data were analyzed descriptively (proportions, median, mean, and standard deviation). The Mann–Whitney test was used to compare the median (interquartile range) between the two groups (grand multipara vs. multipara). Proportions were compared by chi-square between the different groups where appropriate. Any variables with a *p*-value <0.25 in the univariate analysis were further analyzed by logistic regression. The cut-off *p*-value (<0.25) point was taken because its insignificance in univariate and significance in multivariate analysis, and vice versa, is not uncommon [19]. Multivariable logistic regression was performed using grand multipara as the dependent variable and sociodemographic (e.g., maternal age, parents’ education, occupation, child gender, and income rate) and obstetric factors (e.g., mode of delivery, child weight, and gestational age at delivery) were considered as the independent variables. Adjusted Odds Ratio (AOR) and 95% confidence interval (CI) were calculated. A *p*-value of <0.05 was considered statistically significant. The results were illustrated in the text, figure, and tables by calculating the means (SD) and median (interquartile range) for continuous variables, frequencies, and proportions for categorical variables to describe the participants’ responses.

## 3. Results

From the total of 1818 enrolled mothers with complete data, the totals of primipara, multipara, and grand multipara numbered 649 (35.7%, 95% CI 33.5, 37.9), 1034 (56.9%, 95% CI 54.6, 59.2), and 135 (7.4%, 95% CI 7.2, 7.6), respectively (Figure 1). 

The mean (SD) of parity was 2.23 (1.24). Table 1 shows the descriptive characteristics of participants (multipara vs. grand multipara).

Of the total of 1065, 737 (69.2%) and 328 (30.8%) were Arab and non-Arab mothers, respectively. Of the total 737Arab mothers, 14.5% did not receive support during pregnancy in terms of breastfeeding compared to non-Arab mothers (4%). Among 121 grand multipara mothers, 87 (71.9%), 24 (19.8%), 10 (8.3%), and 0 (0%) were Emirati mothers, non-Emirati Arab mothers, Asian mothers, and others, respectively. 

The parents’ education, child’s gender, child’s gestational age at delivery, child’s weight at delivery, and mode of delivery were not associated with grand multipara.

Table 2 shows the unadjusted and adjusted factors associated with grand multipara.

In multivariable logistic regression analysis, factors associated positively with grand multipara were a higher maternal age (AOR = 1.28, 95% CI = 1.21, 1.34), Arab mothers (AOR = 5.66, 95% CI 2.81, 11.40), status as overweight-pregnancy (AOR = 2.01, 95% CI = 1.26, 3.21), and the receipt of less support from family (AOR = 2.05, 95% CI = 1.21, 3.50).

## 4. Discussion

The main key findings of the present study were the estimation of grand multipara and its associated factors, including higher maternal age, Arab mothers, status as overweight pre-pregnancy, and receipt of less support from family. The current prevalence of grand multipara (7.4%) is low compared to previous assessments in the UAE (36.6%) in 2001 [4] and Ethiopia (70.8%) in the demographic and health survey of 2000–2016 [13]. On the other side, a low prevalence of grand multipara was reported in Jordan, at 1.8% [1]. It is clear there is a dramatic drop in the prevalence of grand multipara in the UAE (i.e., from 36.6% to 7.4%). This raises the importance of investigating grand multipara issues thoroughly in terms of all aspects (i.e., culture, health, etc.) and taking appropriate actions by all involved parties. 

There are clearly considerable variations in the prevalence of grand multipara among countries and even within countries overtime. Such variations can be explained by the nature of the studies (i.e., designs and setting). However, the rising global trend of having fewer children is even documented in Africa [2]. Such a phenomenon can be explained by many reasons: high maternal education, late marriage, economic hardship, and the use of different contraception methods. For example, in 2005, Paul et al. reported that daughters were more likely to delay the age of marriage, use contraception methods, and desire to have fewer children in comparison to their mothers in the UAE [10].

In this study, grand multipara mothers were significantly associated with being overweight. Likewise, several studies documented the same results [20,21]. For example, Iversen et al. estimated an average gain of 0.62 BMI units after each additional birth [21]. The authors’ previous published data showed an association between increased parity and weight gain; higher maternal age was associated with overweight and obesity [22]. The combined effects of higher maternal age and higher parity on overweight and obesity could explain the high prevalence of hypertension and diabetes among grand multipara [1,4]. The literature from many countries, including the UAE, revealed that chronic diseases (e.g., diabetes mellitus) are associated with adverse pregnancy outcomes [23,24]. For example, a systematic review and meta-analysis conducted by Relph et al. showed mothers with uncontrolled diabetes mellitus were at greater risk of preeclampsia, preterm birth, perinatal death, congenital abnormality, small for gestational age, and admissions to NICU [23].

The results revealed a highly significant association between higher maternal age and grand multipara. Consistent with the current results, many previous studies reported a similar association [3]. The higher maternal age could have contributed to the poor reported outcomes associated with grand multipara; hence, higher maternal age is associated with several adverse pregnancy outcomes, such as high CS [25]. For instance, the poor obstetrics and perinatal outcomes (e.g., gestational diabetes, hypertension, preterm labor, CS, LBW, and NICU admission) were primarily related to higher maternal age rather than grand multipara among Jordanian mothers [1]. Thus, higher maternal age can be considered the most prominent factor in grand multipara adverse pregnancy outcomes. A similar conclusion regarding the impact of maternal age was also drawn by a study conducted in the neighboring country of Saudi Arabia [9]. 

In contrast to HICs such as the UAE, in Sub-Saharan Africa, young maternal age was a strong predictor for adverse pregnancy outcomes, such as preterm birth and LBW [26]. This could be due to many reasons, such as the low maternal healthcare service utilization in the majority of Sub-Saharan African countries [27], limited access to prenatal care and education among teenage mothers [27,28,29], and as a consequence, early maternal marriage with its adverse pregnancy outcomes. It needs to be mentioned that almost all enrolled mothers in this study were 18 years old and above, and the majority (95.5%) were educated (secondary education and above).

Interestingly, although Arab mothers were almost six times more likely to be grand multipara compared to their non-Arab counterparts, there is still an obvious decrease in the parity pattern among Arab mothers. The present results showed a reduction in the grand multipara pattern, even among Arab mothers [4]. Therefore, in descending order, grand multipara was more common in Emirati mothers, non-Emirati Arab mothers, Asian mothers, and others. Such an order of prevalence of grand multipara can be explained by many factors; of them, the status as a resident (the UAE) is the most important. Giving expatriates an extended stay visa (Golden Visa) could change the prevalence of grand multipara. This confirms the authors’ assumption of the need for future research to update the current data in such a dynamic country. 

Interestingly, the current results showed that none of the mothers from Western countries had five deliveries or more (grand multipara). This can be explained by the early introduction of family planning in Western countries [11], in addition to status as a resident (the UAE).

The present data showed that grand multipara, especially Arab mothers, were less likely to receive support for breastfeeding from family members compared to multipara counterparts. This indicates that the associated factors of grand multipara are interlinked (i.e., national and support) as both nationality and receiving support were associated with grand multipara. In the presence of the higher risk of grand multipara during and after delivery, especially in LMICs, greater support from all family members is recommended for grand multipara mothers aiming to overcome challenges with good pregnancy outcomes.

Delivering prenatal and postpartum breastfeeding support to all mothers, regardless of parity, is highly recommended by researchers to improve breastfeeding outcomes [30]. Nevertheless, parity should be considered when counseling mothers on decisions about infant feeding practices [31].

This study showed that sociodemographic factors were more prominent and associated with grand multipara mothers than obstetrical ones. Unlike previous studies, the current results showed no association between grand multipara and adverse pregnancy outcomes (e.g., cesarean delivery, fetal weight, and preterm birth), which are observed in LMICs [5] such as Sudan and Ethiopia [3,6]. The current results were similar to previous studies conducted in the UAE [4] and Saudi Arabia. For example, in Saudi Arabia, Al-Shaikh et al. encouraged grand multiparity in the presence of the good provision of perinatal care [9].

The low incidences of adverse pregnancy outcomes among women with grand multipara could be explained by the good quality of health care provided to all mothers in the UAE, regardless of parity. This support the UAE policy that aims to increase the population (i.e., encouraging more deliveries with good provision of health care) [4].

The present study has many strengths, such as it tackled a very important topic (parity profile) related to health and culture in multinational country (UAE), and it was conducted in the largest emirate of the UAE (Abu Dhabi) among the seven emirates [32], which made it a great representation of the UAE (these results will act as a basis for further studies). However, there are some limitations that need to be considered. Of them, first, unlike previous studies [2,3,4], not all obstetrical outcomes (such as maternal and perinatal mortality and morbidity) were included, as the current study data were part of a large project. Second, the study design was based on quantitative instruments rather than a study with mixed methods (both quantitative and qualitative), which would better explain such an understudied topic.

Therefore, further studies are recommended to overcome the present limitations and explore the complexity of parity (e.g., the impact of parity on health and culture and vice versa, including the sociocultural aspects). It is worth emphasizing that, in future studies, asking for parity as well as the number of living children is crucial, as it was documented that high parity was associated with high child mortality (i.e., having more births for compensation) [12]. Although the current study explores grand multipara issues (prevalence and sociodemographic profiles of grand multipara) in the UAE and its complexity, it raises new questions that need to be answered by future studies.

Therefore, it is recommended in future studies to include a multidisciplinary team from different ministries and authorities (Ministry of Health, Ministry of Culture, Abu Dhabi Early Childhood Authority) to obtain a complete picture of grand multipara issues. Furthermore, it is recommended that future studies focus on grand multipara mental health, especially in the absence of support, as this study showed. For example, a cohort study from Finland showed maternal grand multipara predicted a significantly increased risk of mood disorders, non-psychotic mood disorders, and suicide attempts among adult offspring [33].

To maximize the benefits of the current study, the authors will communicate the findings of the present results to wider audiences through publication, delivering presentations at conferences and workshops, and sharing the information with the policymakers in the UAE as well as with colleagues.

## 5. Conclusions

The current study provided updated data regarding grand multipara to decision-makers in Abu Dhabi, the UAE. The current prevalence of grand multipara was low compared to previous ones in the UAE. Sociodemographic factors were more prominent and associated with grand multipara mothers compared to obstetrical ones. To support grand multipara mothers aiming to improve maternal and child health, more programs (nutritional, physical activities, psychosocial) are needed. Dealing with grand multipara is a complex issue; therefore, a comprehensive strategy is needed by all involved parties. Further research is required to explore the difference of parity based on nationalities, especially from a sociocultural point of view. 

## Figures and Tables

**Figure 1 nutrients-14-04686-f001:**
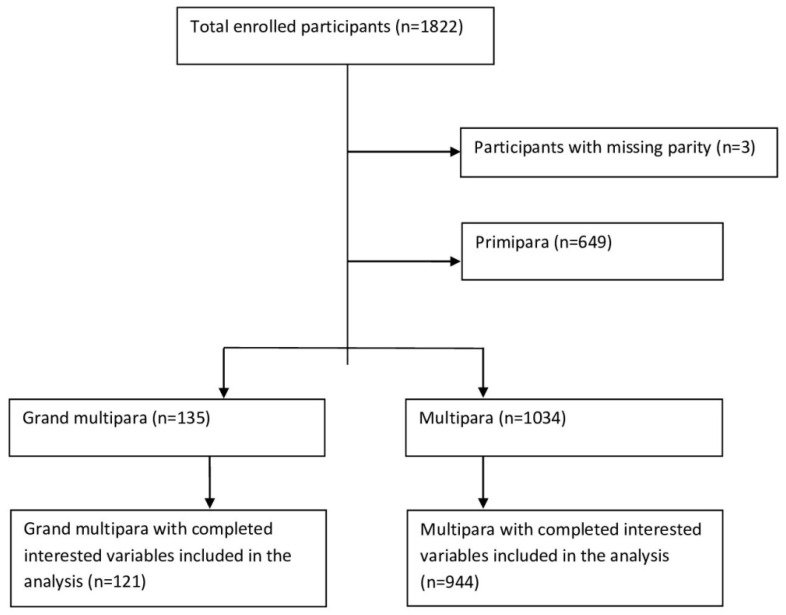
Chart of the selected study participants.

**Table 1 nutrients-14-04686-t001:** Sociodemographic characteristics of the studied participants, Abu Dhabi, United Arab Emirates.

Variable		Total (N = 1065)	Parity		
			Grand Multipara (n = 121)	Multipara (n = 944)	*p*-Value
Median (interquartile range) of					
Maternal age, years		31 (28–35)	37 (34–38)	30 (27–34)	<0.001
Pre-pregnancy BMI		23.8 (21.8–26.3)	26.1 (23.2–28.4)	23.5 (21.7–25.9)	<0.001
		N (%)	N (%)	N (%)	*p*-value
Maternal education	≥Secondary level	1017 (95.5)	113 (93.4)	904 (95.8)	0.236
	<Secondary level	48 (4.5)	8 (6.6)	40 (4.2)	
Paternal education	≥Secondary level	1060 (99.5)	121 (100)	939 (99.5)	0.923
	<Secondary level	5 (0.5)	0 (0.0)	5 (0.5)	
Nationality	Arab	737 (69.2)	111 (91.7)	626 (66.3)	<0.001
	Non-Arab	328 (30.8)	10 (8.3)	318 (33.7)	
Pre-pregnancy BMI	Normal BMI	641 (60.2)	43 (35.5)	598 (63.3)	<0.001
	Underweight	22 (2.1)	2 (1.7)	20 (2.1)	
	Overweight	324 (30.4)	60 (49.6)	264 (28.0)	
	Obese	78 (7.3)	16 (13.2)	62 (26.6)	
Income rating	≥Good	993 (93.2)	109 (90.9)	884 (93.6)	0.142
	<Good	72 (6.8)	12 (9.1)	60 (6.4)	
Received breastfeeding support during pregnancy	Yes	945 (88.7)	90 (74.4)	855 (90.6)	<0.001
	No	120 (11.3)	31 (35.6)	89 (9.4)	
Mode of delivery	Vaginal delivery	749 (70.3)	90 (74.4)	659 (69.8)	0.582
	Planned cesarean delivery	247 (23.2)	24 (19.8)	223 (23.6)	
	Emergency cesarean delivery	69 (6.5)	7 (5.8)	62 (6.6)	
Child gender	Male	505 (47.4)	62 (51.2)	443 (46.9)	0.699
	Female	560 (52.6)	59 (48.8)	501 (53.1)	
Gestational age at delivery	Term (≥37 weeks)	989 (92.9)	110 (90.9)	879 (93.1)	0.375
	Preterm (<37 weeks)	76 (7.1)	11 (9.1)	65 (6.9)	
Child birth weight at delivery	≥2500 g	981 (92.1)	112 (92.6)	869 (92.1)	0.846
	Low birth weight (<2500 g)	84 (7.9)	9 (7.4)	75 (7.9)	
Macrosomia (≥4000 g)	Yes	44 (4.1)	5 (4.1)	39 (4.1)	1.000
	No	1021 (95.9)	116 (95.9)	905 (95.9)	

**Table 2 nutrients-14-04686-t002:** Factors associated with grand multipara using logistic regression in Abu Dhabi, United Arab Emirates.

Variables		Odd Ratio 95% Confidence Interval (CI)	*p*-Value	Adjusted Odd Ratio 95% CI	*p*-Value
Maternal age, years		1.29 (1.22, 1.35)	<0.001	1.28 (1.21, 1.34)	<0.001
Maternal education	≥Secondary level	0.63 (0.29, 1.37)	0.240	0.48 (0.17, 1.40)	0.180
	<Secondary level (reference)				
Nationality	Arab	5.64 (2.91, 10.92	<0.001	5.66 (2.81, 11.40)	<0.001
	Non-Arab reference				
Pre-pregnancy BMI	Normal BMI (reference)				
	Underweight	1.39 (0.32, 6.15)	0.664	0.89 (0.16, 5.12)	0.898
	Overweight	3.16 (2.08, 4.80)	<0.001	2.01 (1.26, 3.21)	0.004
	Obese	3.59 (1.91, 6.74)	<0.001	1.61 (0.78, 3.35)	0.201
Income rating	≥Good	0.62 (0.32, 1.18)	0.145	0.52 (0.24, 1.12)	0.095
	<Good (reference)				
Received support during pregnancy	No	3.31 (2.08, 5.26)	<0.001	2.05 (1.21, 3.50)	0.008
	Yes (reference)				

## Data Availability

The data supporting the current study’s findings are available from the corresponding author upon reasonable request.

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
