# Peer review of "Prevalence and Sociodemographic Profiles of Grand Multipara in Abu Dhabi, United Arab Emirates"

_nutrients, 2022, doi:10.3390/nu14214686_

Round 1

Reviewer 1 Report

Journal: Nutrients (ISSN 2072-6643)

Manuscript ID: nutrients-1988805

Type: Article

Title: Prevalence and sociodemographic profiles of grand multipara in Abu Dhabi, UnitedArab Emirates

Authors: Zainab Taha * , Farid EL Ktaibi , Aysha Al Dhaheri , Dimitrios Papandreou , Ahmed Ali Hassan

Section: Pediatric Nutrition

Revision:

The Article entitled: "Prevalence and sociodemographic profiles of grand multipara in Abu Dhabi, UnitedArab Emirates" the article  aimed to investigate the prevalence and factors associated with multipara mothers compared to multipara mothers in Abu Dhabi, UAE. The study is current and interesting, and the authors take into account several variables. The statistical analysis is appropriate. I think the study is clear and supported by great good results. For the future I would also consider the variable eating habits.

·       The references are in small numbers. Probably because there are few studies about it? If it were possible replenishing the introduction. If it were possible I would replenish the Introduction to add more references.

Reviewer 2 Report

The manuscript by Taha et al presents a descriptive analysis of the prevalence and characteristic of grand multipara in a sample of mothers from UAE. Authors found a 7.4% prevalence of mothers with five or more births, defined as >22 weeks of gestation. A comparison is made with previous data reported for the same country and region.

The authors conclude that the prevalence of grand multipara has decreased in UAE with a difference of approximately 30 percent points (37 to 7%) in 20 years (2001-present). Considering demographic findings better programs directed to this population will help to support parity.

The manuscript is well written, some observations to the format were found, see below.

Section 2. Materials and Methods appears to be copied from the journal’s instructions, please delete.

Define CS at first mention

Line 215 and 228 “other” studies, change it for “previous” studies

Reference list needs to be revised, several references are incomplete or show mistakes, for example:

Ref 2 Muniro, Z., Tarimo, C.S., Mahande, M.J. et al. Grand multiparity as a predictor of adverse pregnancy outcome among women who delivered at a tertiary hospital in Northern Tanzania. BMC Pregnancy Childbirth 19, 222 (2019). https://doi.org/10.1186/s12884-019-2377-5

Ref 8 Nargund G. Declining birth rate in Developed Countries: A radical policy re-think is required. Facts Views Vis Obgyn. 2009;1(3):191-3. PMID: 25489464; PMCID: PMC4255510.

Ref 6 Int J ofWomen’s Heal.
